# Sensuator: A Hybrid Sensor–Actuator Approach to Soft Robotic Proprioception Using Recurrent Neural Networks

**Pornthep Preechayasomboon** * and **Eric Rombokas**

Mechanical Engineering, University of Washington, Seattle, WA 98195, USA; rombokas@uw.edu
* Correspondence: prnthp@uw.edu

**Abstract:** Soft robotic actuators are now being used in practical applications; however, they are often limited to open-loop control that relies on the inherent compliance of the actuator. Achieving human-like manipulation and grasping with soft robotic actuators requires at least some form of sensing, which often comes at the cost of complex fabrication and purposefully built sensor structures. In this paper, we utilize the actuating fluid itself as a sensing medium to achieve high-fidelity proprioception in a soft actuator. As our sensors are somewhat unstructured, their readings are difficult to interpret using linear models. We therefore present a proof of concept of a method for deriving the pose of the soft actuator using recurrent neural networks. We present the experimental setup and our learned state estimator to show that our method is viable for achieving proprioception and is also robust to common sensor failures.

**Keywords:** soft robotics; soft actuators; soft sensors; neural networks; deep learning; state estimation

## 1. Introduction

Progress in soft robotics research is accelerating, and soft robots are impacting a variety of practical applications, from grasping to search-and-rescue. The steady advances in this area have generally focused on integrating physical intelligence for robust and useful open-loop behavior [1,2]. By taking advantage of characteristics such as compliance, passive stability and flexibility, the advantages of these robots have mostly been realized using only limited sensing. We expect continued progress in this domain, but soft sensors are a key capability for realizing the full potential of soft robots. Soft sensors are challenging to design and fabricate, and it can be difficult to interpret their outputs. Despite these challenges, significant progress has been made using a variety of sensor modalities including piezoelectric tactile sensing [3,4], resistive [5,6], capacitive [7] and inductive [8] stretch sensors, magnetic effects [9] and optical waveguides [10–12]. Integrating soft sensors relies on more than taking advantage of various modalities; the sensors must be physically coupled to the robot in useful ways, and the data must be useful for planning or control.

Ideally, we could design and fabricate highly sensorized soft robots to rival the sensor density and acuity found in the animal world. However, dedicated sensor structures add fabrication complexity and require physical space. Biologically inspired sensor strategies would ideally include dense, diverse sensors in currently unrealizable quantities. For instance, the fingertip contains an estimated two to four sensory organs for sensing touch per square millimeter [13]. Multimaterial 3D printing has enabled the fabrication of embedded sensors using conductive ink for touch, pressure, motion and temperature sensing [14,15], which is somewhat similar to the multi-modal nature of biological systems. Recent work, drawing inspiration from the human perceptual system, has demonstrated that a redundant, unstructured sensor topology may be modeled using modern neural networks. This approach enabled force and displacement models for a soft robot that were robust to sensor non-linearities and drift [16]. To date, the aforementioned approaches have relied on purposefully built sensor structures within the soft robot and may require

the use of machine learning to fully utilize the sensor responses. In a similar vein, with the non-linear behavior surrounding the materials and unstructured nature of soft robotics, various facets of soft robotics research have turned to the use of data-driven or deep learning based methods for tasks such as design and fabrication [17], sensing [18,19], state estimation [20,21] and control [22–24].

In this proof-of-concept paper, we present a hybrid sensor–actuator design that, instead of relying on purposefully built sensor structures, takes advantage of the actuated fluidic medium of the robot to perform a double duty as a sensing medium in a bellows-style soft robotic actuator. Prior work by Helps and Rossiter [25] has demonstrated the use of salt water as both an actuated and sensing medium to derive the bending angle of a bellows-style actuator and closed-loop control. That system used a single sensor to estimate the whole-robot bending angle. In contrast, our approach relies on a distributed network of sensors that are created by the volume and geometry of salt water in each bellow. We also use flexible conductive silicone electrode segments that more closely match the material properties of the robot. Due to the unstructured nature of our sensors, we rely on a data-driven model, a recurrent neural network, for proprioceptive state estimation. The multi-channel sensor-actuator provides a six-dimensional space that captures both the tip location and high-fidelity pose. We present the design, experimental setup and state estimation examples of the sensor–actuator hybrid as well as a brief investigation into common sensor failure modes and an exploratory machine learning architecture investigation for our data.

## 2. Materials and Methods

In this section, we provide details on how we fabricated our hybrid sensor–actuator, as well as an experimental setup for validating our approach. Our soft robotic actuator, as shown in Figure 1, was a bellows-style actuator similar to the PneuNet actuator [2], where, upon an increase of pressure inside the bellows, the actuator would bend. Each bellow had a conductive silicone tab on each side that could conduct electricity from the internal fluid to the outside; thus, each bellow formed a resistor when a conductive fluid was used. There were seven bellows in total, and the actuator was approximately 25 mm by 26 mm by 90 mm in size. To track the actuator's position in space, we attached seven infrared (850 nm wavelength) LEDs underneath each bellow.

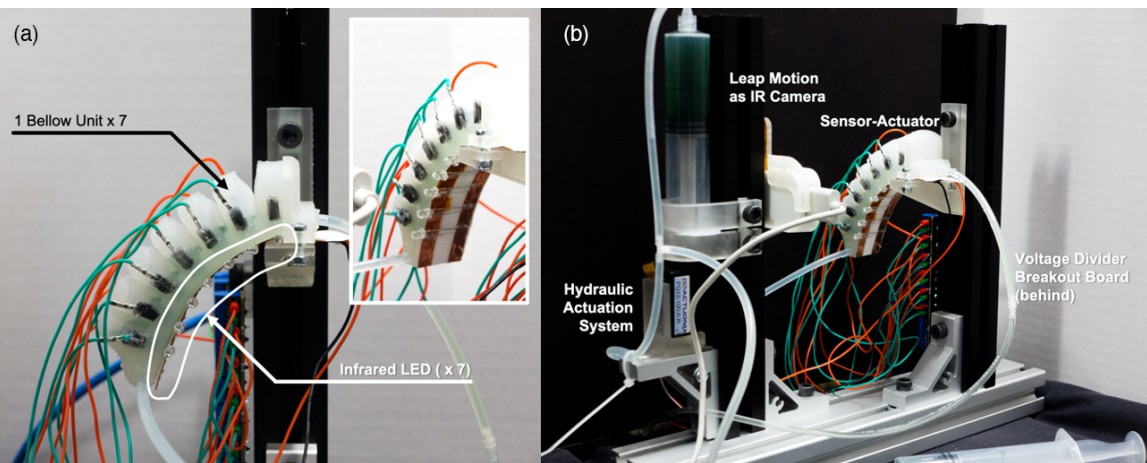

**Figure 1.** (**a**): Our soft robotic sensor–actuator in its idle pose. Shown in the inset are the seven infrared LEDs on a copper strip attached to the actuator, which also acts as a strain-limiting layer. (**b**): The experimental setup with the hydraulic actuation system is shown, along with the Leap Motion infrared camera and breakout board for data acquisition.

### 2.1. Actuator Fabrication

Our bellows sensor–actuator hybrid was fabricated primarily using the Negshell casting method [26]. Negshell casting is a sacrificial casting method that utilizes the

inherent fragility of stereolithography-printed (SLA) resins to create breakable cores that become hollow structures inside soft robots after casting. We first 3D printed the internal sacrificial cores and external molds for our actuator using clear resin on a Formlabs Form 2 SLA printer. The molds had 8 by 4 mm openings along the side of each bellow chamber that could be removed after casting—these openings were later used to house conductive silicone. Then, we prepared 60 g of PlatSil Gel-25 silicone elastomer (Polytek Development Corp., Easton, PA, USA) by mixing equal parts of Part A and Part B, followed by vacuum degassing. The mixture was injected into the mold with a syringe and left to cure for 12 h. After the silicone elastomer had set, conductive silicone, as explained in [27], was casted into the openings on the side of each bellow. After the conductive silicone had set, the sacrificial Negshell cores were crushed by hand to create flexible inner chambers. Then, 3 mm diameter flexible silicone tubes were attached using Sil-Poxy (Smooth-On, Inc., Macungie, PA, USA) and any leaks were patched using the same adhesive. Lastly, 28 AWG silicone-sheathed electrical wires were pierced into the conductive silicone tabs to create an electrical connection. The actuator was filled with a 5 wt% sodium chloride (table salt) solution mixed with green food coloring.

### 2.2. Experimental Setup

We built an apparatus to actuate the sensor–actuator using aluminum extrusions as a frame, as shown in Figure 1. The sensor–actuator's base was secured onto a 3D-printed pedestal attached to the vertical extrusion. A syringe pump constructed from a linear actuator servo (PQ12-R, Actuonix Motion Devices Inc, Saanichton, BC, Canada.), a 30 mL syringe and 3D-printed parts were attached and connected to the actuator with silicone tubing. A hand-actuated 30 mL syringe was also hydraulically connected in parallel to the servo-actuated syringe. To track the positions of the infrared LEDs, we attached a Leap Motion Controller (Ultraleap), used as an infrared camera with a global shutter, opposite from the actuator. A silicone tube from the actuator was also connected to a pressure transducer (ABPMANT100PG2A3, Honeywell International Inc., Charlotte, NC, USA).

To measure the conductance of each bellow, we used the method outlined in [27], where a sinusoidal waveform of 2 volt peak-to-peak centered at 0 volts and a frequency of 100 Hz was applied across each bellow and the voltage drop across the resistors was measured, as shown in Figure 2. We used resistors with a resistance of 56 kΩ as the fixed resistor for each bellow. As the shape of the bellows changed, the resistance across the bellows would also change, which can be loosely modeled using the law of resistance ($R = \rho \frac{L}{A}$). An NI USB-6251 (National Instruments) data acquisition system was used to generate the waveform as well as to measure the voltage across each bellow.

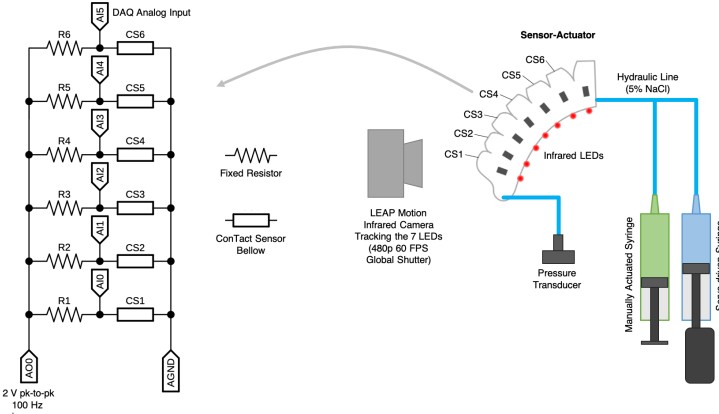

**Figure 2.** Soft hydraulic robotic finger system diagram using hybrid sensor–actuators. Each of the sensing elements (CS1-CS6) spans a bellow of the hydraulic bellows-style actuator. A series of infrared LEDs provides an external "ground truth" of the robot pose for training and performance quantification.

### 2.3. Data Collection

The entire aforementioned system was controlled using a PC running LabVIEW (National Instruments) and a Python script. Our Python script processed the image from the left camera of the Leap Motion Controller using the OpenCV library to segment and track the positions of the seven infrared LEDs at 50 Hz. Our LabVIEW application collected the voltage measurements across each bellow along with pressure. The syringe driver's linear servo was controlled using pulse-width modulation (PWM) as the commanded stroke length, which was randomly generated every 1 s. The collected pressure and voltages were sent to the Python script via the local network to be logged to a comma-separated values (CSV) file, which was timestamped by each camera frame.

We collected two sessions of "babbling", where the actuator was randomly set to different pressures using both the linear servo and a hand-actuated syringe. Throughout each session, we used a hand to block or disturb the actuator at random. The first session lasted approximately 12.5 min or 37,336 samples and served as the training dataset for our neural network. The second session lasted 10 min or 30,000 samples and served as the test dataset for our neural network. A sample video during data collection is provided in the Supplementary Materials.

### 2.4. Machine Learning Models

With the popularity of deep learning and research surrounding the topic, deep learning has transitioned from being a research topic to becoming a tool to enable novel methods in other research areas. In our case, our sensor–actuator's sensor data had several unwanted properties that would lead classical control schemes or models to fail, the most prominent being the non-linearity and coupled interaction between the resistance-based sensors connected electrically in parallel. Thus, using a linear model to derive the actuator's pose from the sensor data was not an option. We therefore turned to deep learning approaches. We chose a recurrent neural network, the bidirectional variant of Long Short-Term Memory (BiLSTM), for our approach. As previously mentioned, research aiming to provide proprioception for soft robots has used recurrent neural networks [16,20]. LSTM-based models are well suited for time-series predictions and excel in situations where the data are noisy, have time-lagged dependencies and hysteresis [28]. The bidirectional variant of LSTM is fed both data from the past and future during training; thus, BiLSTM is thought to be able to predict future states with higher accuracy due to the better context. We used MATLAB's Deep Learning Toolbox for training and testing our deep learning model, and the relevant hyperparameters are presented in Table 1.

MATLAB's Deep Learning Toolbox consists of a set of functions for creating, training, evaluating and using deep learning models. To construct the neural network for our BiLSTM network, we created our input layer using `sequenceInputLayer()`, followed by a `bilstmLayer()` layer for our BiLSTM layer, and finally a `fullyConnectedLayer()` layer for our fully connected layer. Other architectures could be constructed in the same fashion. A graphical tool, the Deep Network Designer, can also be used to export networks as `.mat` files to be loaded as a single `struct` type variable. After loading our data into memory and a performing pre-processing, we used the function `trainNetwork()` to train our model. Finally, to use the trained model for prediction, we used the `predictAndUpdateState()` function. A MATLAB LiveScript providing step-by-step instructions to train and evaluate our model is available in a repository linked in the Supplementary Materials section. Our raw data is also provided in the same repository.

**Table 1.** Bidirectional Long Short-Term Memory (BiLSTM) model properties and hyperparameters. RMSE: root mean square error.

| Parameter | Value | Notes |
| --- | --- | --- |
| Inputs | 7 | $1 \times$ Pressure, $6 \times$ ConTact, z-score normalized |
| Outputs | 14 | X and Y location of infrared LEDs in pixels, z-score normalized |
| Number of Hidden Units | 100 | - |
| Mini-batch Size | 512 | - |
| Initial Learning Rate | 0.01 | - |
| Learning Rate Drop Period | 5 | Drop by factor every 5 iterations |
| Learning Rate Drop Factor | 0.5 | Next iteration learning rate = $0.5 \times$ previous |
| Optimizer | Adam | - |
| Gradient Clipping | 10 | - |
| Iterations | 50 | - |
| Final Training RMSE | 4.44 | in millimeters across all LEDs |
| Final Test RMSE | 5.85 | in millimeters across all LEDs |

## 3. Results

In this section, we show how our hybrid sensor–actuator can provide rich and informative signals that can be interpreted by learned models. The models were capable of creating reasonable estimates of the positions of the LEDs given time series of sensor data. After training our BiLSTM model, we used it to create time series predictions using our test data as the input. The test data were z-score normalized using the mean and variance for each input from the training data. We describe the preferred estimator's performance in achieving state estimation, examine how sensor degradation affected that performance and compare how different machine learning model choices behaved under sensor degradation.

### 3.1. State Estimation

Over our entire test dataset, which spanned 30,000 samples or 10 min, we achieved a root mean square error (RMSE) of 5.85 mm considering the positions of all seven infrared LEDs. Figure 3 shows the entire test dataset along with several snapshots of the actuator's pose. For simplicity, only the position of the fingertip is plotted, ranging approximately 60 mm in the horizontal (X) direction and 40 mm in the vertical (Y) direction. We observe that the model captured the overall pose of the actuator given sensor inputs throughout the time shown. By inspecting the pose snapshots (Figure 3 below), we see that at moments where the actuator's tip diverged from the ground truth, the overall pose was still mostly correct. The model also responded well to rapid changes, shown around the 100 to 200 s mark. This result shows that the BiLSTM model generally captured the actuator's nonlinear response and time-dependent characteristics well. Videos of the resulting poses are available in the Supplementary Materials section.

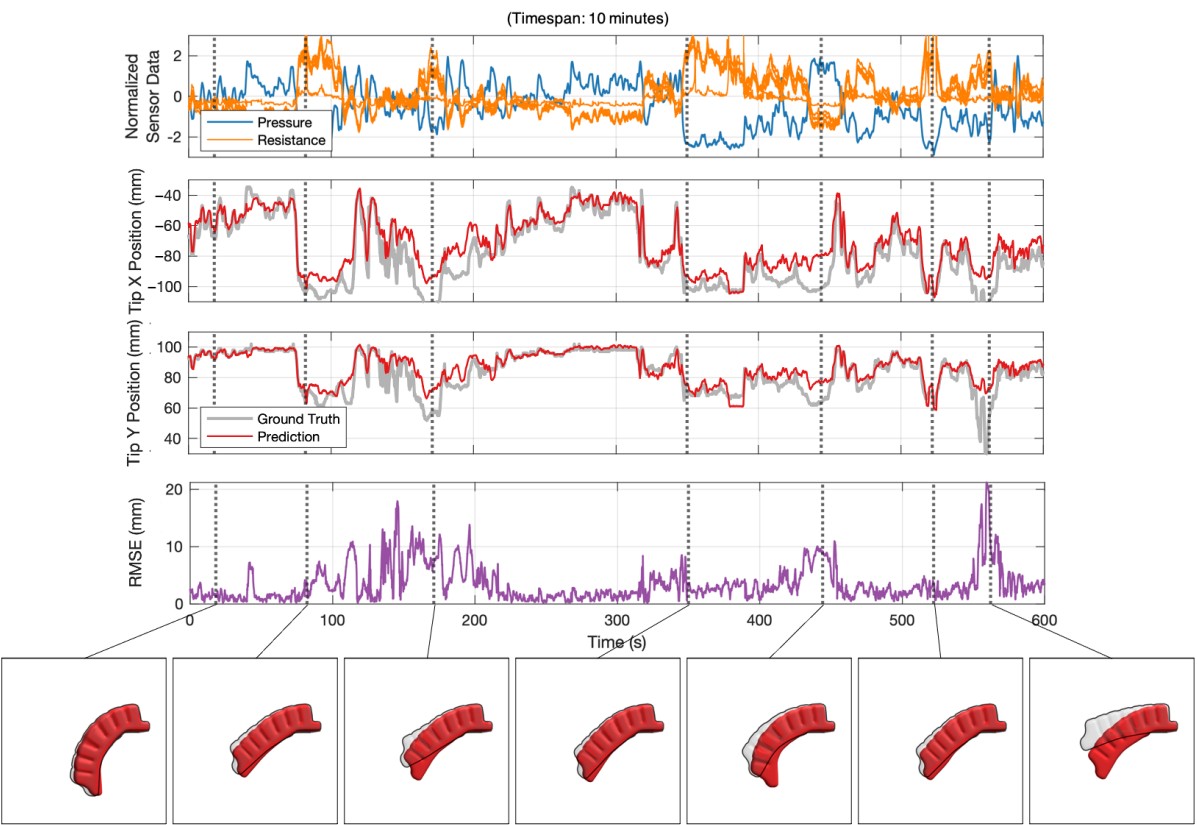

**Figure 3.** The entire 10 min of the test dataset is shown along with predictions from the BiLSTM model and snapshots of the resulting poses. The top row depicts the raw sensor data, showing both the single pressure sensor and the suite of resistances. The second and third rows show only the fingertip horizontal (X) and vertical (Y) positions. The fourth row is a plot of the root mean squared error across all the LED positions. The insets below depict the overall pose of the robot during seven moments in which the pose estimator was performing maximally or minimally well.

### 3.2. Sensor Degradation

In practice, sensors can degrade over time or fail completely. In biological systems, it is possible to compensate for or adapt to a loss of sensory feedback. Since our sensors may or may not have redundancy, we aimed to investigate the robustness of our BiLSTM model towards the loss of input data. Here, we attempted to artificially degrade the sensors by degrading the inputs from the test dataset and evaluated the resulting predictions from the previously trained model. In Figure 4, we present 4 min excerpts of the resulting outputs of the trained model generated from input data that were gradually degraded by artificially saturating the inputs in turn. We saturated the sensors by setting the entire duration of the sensor's input to zero before z-score normalization. In the first case, we completely saturated the first two resistance-based sensors near the tip. The plot shows that, in general, the model still performed well. In the case with four sensors saturated, the model still predicted the movement well, but with an expected overall decrease in accuracy. However, with six sensors saturated, or with only the pressure readings intact, the model performed extremely poorly. It is notable that some dynamics were still captured using pressure alone, suggesting that the model leverages pressure readings to some extent. Conversely, with only the pressure readings removed and leaving all other sensors intact, the model performed considerably well. Arguably, this result shows that the input data may not be redundant after all, but this is difficult to determine due to the black-box nature of neural networks.

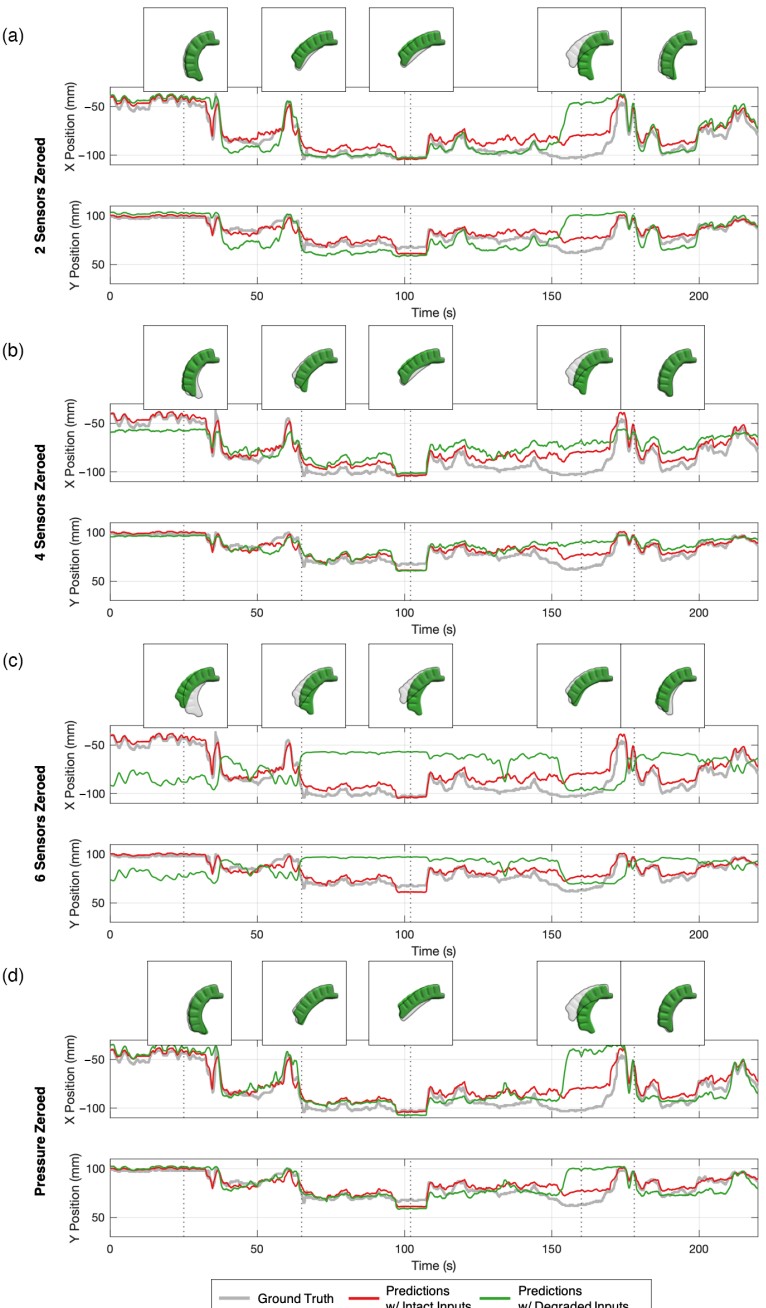

**Figure 4.** Plots of the tip position in X and Y in mm and snapshots of the pose of the sensor–actuator when estimated under various states of sensor degradation: (**a**) the voltage readings from the resistance-based sensors when the first two bellows from the tip of the sensor-actuator are saturated, (**b**) the first four sensors from the tip are saturated, (**c**) all six resistance-based sensors are saturated, leaving only the pressure reading intact, and (**d**) only the pressure sensor is saturated.

*3.3. Network Architecture Choices*

We also investigated other deep learning architectures that have been commonly suggested in the literature for time-series data and non-linear sensors, namely multi-layer perceptron (MLP), gated recurrent units (GRU) and long short-term memory (LSTM). To test the robustness of these models towards sensor degradation and compare them to BiLSTM, we first trained each respective model with clean input data from the training dataset; then, the resulting estimations from both clean and degraded input data from the test dataset were compared. We degraded the test data in several ways simultaneously: (1) the pressure data were saturated, (2) Gaussian noise was applied to one resistance-based

sensor, and (3) the third resistance-based sensor from the tip was constantly saturated. The results shown in Figure 5 show that MLP fared worst out of all models tested, where Gaussian noise overwhelmed all outputs. GRU and LSTM-based models were more robust to noise but lost a considerable amount of accuracy, further solidifying our choice of using BiLSTM as our neural-network architecture.

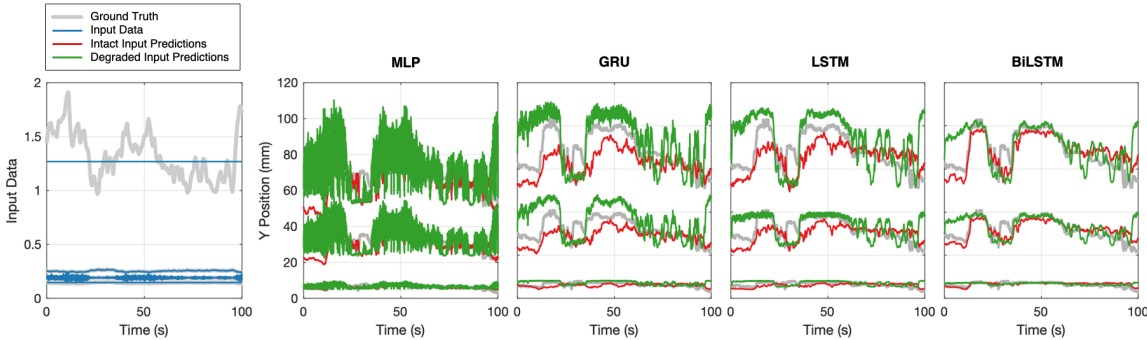

**Figure 5.** Results from comparing across models with the following conditions: (1) pressure data were constantly saturated, (2) Gaussian noise was multiplied to the data from sensor 2, and (3) sensor 3 was constantly saturated.

## 4. Discussion

We have presented a proof-of-concept towards regarding the fabrication of a soft robotic actuator that does not require purposefully-built sensor structures. We show that our sensor–actuator hybrid can provide relatively accurate estimates of its pose despite input signals that originate from unstructured, noisy and time-lagged sensors. For many applications, it is reasonable to simply construct distinct, dedicated sensor structures alongside the actuator structures. A variety of research has demonstrated state estimation and closed-loop control using that approach (e.g., [15]). However, as the scale of the desired robot becomes smaller and as the number of desired sensors and actuators increases, simplicity and compactness become increasingly important. Combined sensor-actuators could serve as a strategy for dense and small sensors, with distinct engineering tradeoffs.

Sensors, especially in robotics, are commonly designed to have well-defined properties that are linear, repeatable and accurate. Most real-world sensors do not exhibit such behaviors under wide conditions, and those sensors that can be made to exhibit those characteristics require careful engineering and are often expensive. In biological systems, sensors are magnitudes worse than man-made sensors in most aspects, such as hysteresis and dependence on temperature, yet humans can perform intricate tasks in the noisiest of environments.

A key question for soft robotic sensing, then, is the degree to which the sensor interpretations should be based on an a priori model vs. empirical data and function approximation [29]. In this paper, we embrace the shortcomings of our sensors and pursue an empirical, data-driven approach. Our method is only one of a class of methods for using machine learning to make the most of complex soft robotic sensors [30,31]. We use neural networks, which can be extremely powerful for time series regression and forward prediction tasks. However, as with all empirical techniques, they come at the cost of requiring a large number of examples along with relatively clean ground truth data. Their performance is not guaranteed to extrapolate data when starved of examples in particular scenarios—an example is shown in the last snapshot in Figure 3, where the particular pose was not present in the training data. However, through the sensor degradation tests, we show robustness to other common types of faults.

In terms of practicality, our presented prototype still has several limitations. Firstly, using salt water as the actuation fluid increases the weight of the actuator greatly, especially when compared to its pneumatic equivalent. We hope to mitigate this in future prototypes by using the actuation fluid more effectively and putting emphasis on the design of the bellows themselves, such as the miniaturized bellows shown in [26]. The salt

water solution used in our experiments is cheap and easy to prepare but is susceptible to evaporation due to the permeability of silicone rubber. This can be mitigated by using other solvents, as presented in other research [32]. Nevertheless, we believe that our methods are easily transferable to other solvents and actuator designs that use ionic liquids as the actuation medium.

Another limitation of the apparatus and experiment we show here is that it does not examine the effects of long-term cycling and time on the sensor performance. However, changes in sensor properties due to use do not appear to have a degrading effect during this experiment. Changes in the strain-resistance properties appear to be negligible or are perhaps accounted for by the machine learning model. This is demonstrated by the similar performance of the estimator at the beginning of the experiment and towards the end, as shown in Figure 3. Over time, the model predictions still hold with a minimal change in the overall pose output.

Hybrid sensor–actuators, combined with machine learning for signal interpretation, offer a path toward sensorized soft robots while adding minimal additional fabrication complexity. With this in mind, our immediate next step for future work will be to apply our technique for the closed-loop control of a soft sensor–actuator. Some challenges still need to be met, such as the time-lagged response to hydraulic actuation and the reduction in weight of the sensor–actuator. We believe that, at the least, our work serves as a proof-of-concept of our approach and will inform the selection of neural-network models for future studies related to soft robotic sensing and actuation.

**Supplementary Materials:** The following are available online at https://www.mdpi.com/2076-0825/10/2/30/s1. Video S1: Training data predictions at real-time speed, Video S2: Training data predictions at 10x speed, Video S3: Test data predictions at real-time, Video S4: Test data predictions at 10x speed, Video S5: Sample video of data collection. Data S6: The MATLAB scripts and raw data used in this paper.

**Author Contributions:** Conceptualization, P.P. and E.R.; methodology, P.P.; software, P.P.; validation, P.P.; writing—original draft preparation, P.P. and E.R.; writing—review and editing, P.P. and E.R.; visualization, P.P.; supervision, E.R. All authors have read and agreed to the published version of the manuscript.

**Funding:** This research received no external funding.

**Acknowledgments:** We would like to thank the members of Rombolabs for their valuable input.

**Conflicts of Interest:** The authors declare no conflict of interest.

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
