# Peer review of "Sensuator: A Hybrid Sensor–Actuator Approach to Soft Robotic Proprioception Using Recurrent Neural Networks"

_actuators, doi:10.3390/act10020030_

Round 1
Reviewer 1 Report
Dear editor, the authors have failed to do a comprehensive literature review. A lot of the works have already initiated in the application of closed-loop control of soft robots and incorporation of NN in modeling and control of them. However, the authors seem not to be aware of them. Also, the methodology is not justified particularly in the use of this type of NN as against others. All in all, the study is not acceptable in the current form for publication.
Best regards.
Author Response
Dear Reviewer 1,
Thank you for the review.
- To address your concern about our literature review, we have added numerous citations for the use of neural networks and deep learning in the sensing, modeling, and control of soft robots to our manuscript’s introduction. We have also reworked the introduction, explaining and emphasizing that our proof of concept is in the domain of a hybrid sensor-actuator design, a distinct concept from simply including sensors in closed-loop control of a soft robot.
- As for the particular neural network architecture choice, the manuscript describes the reasoning for choosing (Bi)LSTM in section 2.3. To enhance and improve this justification, we have added citations from other papers regarding the use of recurrent neural networks. Furthermore, in section 3.3, we investigated the use of other neural network architectures and compare their performance. That comparison specifically addresses the point of justification of the use of this type of neural network.
Reviewer 2 Report
This is a well-organized and articulated manuscript, describing an interesting intersection between soft robotics and neural networks. The authors have motivated their concept logically and soundly in the Introduction, and have included detailed experimental details.
High-level questions and suggestions:
- Do you have an intended use case for your actuator?
- Following on to (1), do you foresee any risks associated with the construction or operation of your actuator in this use case?
- Have you investigated the performance of your actuator and sensing system over many cycles? I suggest a cyclic bending test with 50-100 contractions.
- In Section 3.2, can you explain why the model over- or under- estimates the ground truth for various actuator poses?
- It would be interesting to more closely correlate sensor-actuator performance with specific, pre-defined poses. Perhaps this would provide more granular understanding of why the model fails in certain positions when various sensors are zeroed, which motions perform best/worst, and/or how the system may be improved to be more robust in these situations.
Author Response
Dear Reviewer 2,
Thank you for the review and insightful suggestions.
We would like to answer questions one-by-one:
- We have framed our work as a proof-of-concept (thus the communications submission rather than an article), however, our immediate future work will focus on reducing the “clunkiness” of the overall actuator to achieve a more practical embodiment of the concept presented, such as a high degree of freedom actuator for dexterous grasp and manipulation.
- We have included additional text in the discussion outlining shortcomings from using salt water as the actuation fluid. One of the most obvious problems is how liquid adds to the overall weight of the actuator. We hope to mitigate this using more efficient bellow designs such as those that require minimal volume during its idle state (an example is the design presented in our previous publication: Negshell casting: 3D-printed structured and sacrificial cores for soft robot fabrication).
As also addressed in our discussion, there are risks associated with deploying a trained model in real-world situations, where unforeseen/untrained external disturbances may adversely affect the model outputs/predictions. We would need to further characterize and capture more training data to address these risks. - We have updated Figure 3 to include the entire 10 minutes of our test dataset. Although we do not explicitly capture repeated cyclic actuation, we believe that our randomized actuation is more akin to that of a biomimetic system. The focus of our paper is enabling proprioception in an actuator rather than the actuator’s performance itself.
- & 5. After careful analyses from previously captured video, we have found that most poses where the model fails are due to the pose being never captured in the training data - this highlights a key challenge in deep learning where the training data is insufficient for the model or the model fails to extrapolate from the given input. We have added text to highlight this issue further in our discussion: “ an example is shown in the last snapshot in Fig.3 where the particular pose was not present in the training data.”
Reviewer 3 Report
To authors,
Combining low-cost sensing material with recurrent neural networks offers a promising route to closed-loop control of soft actuators. This strategy is widely applicable to a diverse variety of actuators driven by fluid, air, heat, and ion exchange. This work is a small and clunky step along a promising direction. Presently, however, it does suffer several major weakness in terms of experiment design. Authors should address following issues either by additional tests or in-manu explanation before it can be considered publishable.
- Dexterous motion requires accurate, repeatable and predictable control of actuators. Using conductive silicone and LED-assisted calibration cannot fulfil the need for accuracy here because strain-resistance correlation of the conducting material is not stable and may vary after many actuation cycles. No cycling test data is provided in the current manuscript, and the only several cycles in Fig. 3-4 cannot support repeatability of sensing accuracy.
- Fig.3 and Fig.4 do confirm the applicability of bellow sensors. However, the system doesn’t demonstrate any self-awareness of strain error or self-adaptive capability to achieve a pre-set deformation.
- Fig.5 compares different models, and the result is of wide interest. Importantly, accuracy and fidelity of sensing data used to empirically train the network is of pivotal significance. Authors therefore should always make sure each model is trained by consistent data, collected from an actuator pre-coated with freshly silicone, so the result is more comparable with each other.
Author Response
Dear Reviewer 3,
Thank you for the review and meaningful insights.
We would like to address the comments point-by-point:
- We agree that this experiment is not designed to fully characterize the stability of the sensor-actuators over long-term cycling. However, changes in sensor properties due to use do not appear to have a degrading effect during this experiment. Changes in strain-resistance properties appear to be negligible or accounted for by the machine learning model. This is demonstrated by the similar performance of the estimator at the beginning of the experiment and the end. We have updated Figure 3 in the manuscript to include the entire 10 minutes of our test dataset that shows that, over time, the model predictions still hold with minimal change in the overall pose output. We have also added text to the discussion acknowledging the limitation of this experiment in terms of characterizing changes in sensor properties over cycles and time.
- We agree that this experiment doesn’t include closed-loop control. This manuscript is intended to showcase the design methodology of the hybrid sensor-actuator idea, and to characterize how well the strategy can be used to achieve state estimation. We intend future experiments to incorporate closed-loop control using these sensors, but we feel that the novel design should be cleanly described here first.
- In section 3.3 and figure 5, we use the same (previously captured) data from the previous sections for each model, therefore the fidelity of input data between the comparisons is preserved and the results are truly comparable. We have added text to clarify how we performed the experiments.
Reviewer 4 Report
The manuscript presents an interesting area of soft robotics. - The introduction need to be extended to include more research about the bending actuators and other types of soft sensors. - please validate the results by compare them with literature.
Author Response
Dear Reviewer 4,
Thank you for the review.
We have extended the introduction to include more research about closed-loop control of soft actuators.
As for validation, in section 3.2, we briefly compare our choice of the BiLSTM model against other models used in literature. For the sake of clarity, we have added citations to papers that have used recurrent neural networks for state estimation. A direct comparison of our design against other soft robotic sensor designs, however, is not included because our sensing modality and the actuator’s response is exclusive to our system. As for comparisons between neural networks that are used for soft robotic proprioception, we have somewhat of a comparison to LSTM, as used by Thurthel et al. [16], as presented in Section 3.3. The results show that BiLSTM performs slightly better than vanilla LSTM. However, our architecture does have the exact same structure since the number of inputs and outputs are not identical.
Round 2
Reviewer 1 Report
I suggest including a step by step guide of using Deep Learning in Matlab in this paper in order to increase the reproducibility of the manuscript and the contribution to the students and new researchers in the field. Since the main topic of the paper is how to apply the Deep Learning tool in soft robotics. This could be explained in the main manuscript and guide through the supplementary files for more details.
Author Response
Dear Reviewer 1,
Thank you for the comments. We think a step by step guide is outside the scope of the paper and would add unnecessary bulk to the manuscript. However, we do agree that our work should be reproducible to its fullest. We have added a paragraph of an overview of the commands used and have included an additional Supplementary Material with a complete end-to-end tutorial of how to preprocess the data, train the network and evaluate it.
You can view the tutorial here: https://sensuator.github.io/SensuatorTutorial.html
And the LiveScript file (and raw data) is available here: https://github.com/sensuator/sensuator.github.io/tree/main/Tutorial
Reviewer 3 Report
Recommend for publication.
Author Response
Thank you
Reviewer 4 Report
No more corrections are required
Author Response
Thank you
Round 3
Reviewer 1 Report
Thanks authors for considering my comments, I hope the manuscript will get the utmost attention in the community now and ready to be published.